# Injectable Antiretroviral Drugs: Back to the Future

**DOI:** 10.3390/v13020228

**Published:** 2021-02-02

**Authors:** Marco Berruti, Niccolò Riccardi, Diana Canetti, Sergio Lo Caputo, Lucia Taramasso, Antonio Di Biagio

**Affiliations:** 1Infectious Diseases Unit, Department of Health Sciences (DISSAL), University of Genoa, 16132 Genoa, Italy; marco.berruti1@gmail.com; 2Department of Infectious-Tropical Diseases and Microbiology, IRCCS Sacro Cuore Don Calabria Hospital, Negrar di Valpolicella, 37024 Verona, Italy; niccolo.riccardi@sacrocuore.it; 3Clinic of Infectious Diseases, IRCCS San Raffaele Scientific Institute, 20097 Milan, Italy; canetti.diana@hsr.it; 4School of Medicine, Vita-Salute San Raffaele University, 20097 Milan, Italy; 5Clinic of Infectious Diseases, Department of Clinical and Experimental Medicine, University of Foggia, 71122 Foggia, Italy; sergiolocaputo@gmail.com; 6Infectious Diseases Unit, Department of Internal Medicine, Ospedale Policlinico San Martino IRCCS, 16132 Genoa, Italy; taramasso.lucia@gmail.com

**Keywords:** ART, injectable, HIV, antiretroviral, treatment, PrEP

## Abstract

Current HIV treatment regimens provide sustained virologic suppression, at least partially restore the immune system and have limited side effects; however, they do not allow viral eradication and they are burdened by daily pill intake with a life-long commitment for the people living with HIV (PHIV). Injectable agents might represent a turning point in the care of PHIV, allowing less frequent administration of antiretroviral treatment (ART), more widespread use of pre-exposure prophylaxis (PrEP) and more stable drug levels in the blood, thus increasing the odds to get closer to end the HIV pandemic. The aim of this manuscript is to give a comprehensive review of injectable antiretrovirals that have been used in the past, which are available now, will be available in the future, and their role in the treatment of HIV infection

## 1. Introduction

About 40 years have passed since the first cases of Acquired Immune Deficiency Syndrome (AIDS) were observed, and the HIV pandemic is still responsible for roughly 38 million people living with the virus and 1.7 million new infections every year [1,2]. Current treatments provide successful virological suppression, at least partially restore immune system and have limited side effects; however, they do not allow viral eradication and they are burdened by daily pill intake with a life-long commitment for the patients [3]. Moreover, in some countries with high HIV prevalence, like in Sub-Saharan Africa, difficulties in anti-retroviral treatment (ART) delivery and poor adherence pose a constant threat to achieve the 90-90-90 World Health Organization (WHO) strategy goals [4,5]. Furthermore, the emergence of multi-drug resistant (MDR) strains can favor virological failure, disease progression, and transmission [6]. Even if only an efficacious and free vaccine may be the answer for HIV eradication, long-acting injectable antiretrovirals may constitute a turning point for better infection control in difficult settings and decrease the emergence of resistance [7,8]. In fact, long-acting ART may improve adherence to therapy and extend prophylactic interventions as pre-exposure prophylaxis (PrEP) to vulnerable populations [9]. The aim of this manuscript is to give a comprehensive review of injectable antiretrovirals that have been used in the past, which are available now, will be available in the future, and their role in the treatment of HIV infection.

## 2. Methods

We performed a MEDLINE/PubMed search on 03/03/2019 and the complete search string was as follows: (((injectable) AND antiretroviral) AND HIV) AND (Clinical trial) AND (“01/01/1990” (Date-Publication): “03/03/2019” (Date-Publication)). The string was then updated on the 30 June 2020 to enclose more recent papers

Of the 361 papers identified, 299 were excluded by title and abstract screening as they were not pertinent to our narrative review. We ultimately selected 36 papers: the full texts and pertinent references were then retrieved and collectively discussed. The decision about inclusion in the present narrative review was ultimately made according to the subjective impression of the authors. Eventually, the review was organized for each drug analyzed in the following paragraphs: (i) “Molecular characteristics”; (ii) “Available formulations”; (iii) “Recommended doses”; and (iv) “Place in therapy”. Finally, a discussion about the future applications of the reviewed drugs is given.

## 3. Zidovudine (AZT)

### 3.1. Molecular Characteristics

Zidovudine (AZT) (3’-azido-2’3’-dideoxythymidine) was the first antiretroviral drug available in the 1980s. Initially studied as an antineoplastic drug, it subsequently showed in vitro and in vivo activity against HIV-1 virus and, in a lesser extent, against HTLV-1 and HIV-2. AZT is a thymidine analogue which intracellularly is phosphorylated to its active 5′-triphosphate metabolite, zidovudine triphosphate (ZDV-TP). The principal mode of action of ZDV-TP is the inhibition of reverse transcriptase (RT) via DNA chain termination after incorporation of the nucleotide analogue. ZDV-TP is a weak inhibitor of the cellular DNA polymerases α and γ and has been reported to be incorporated into the DNA of cells in culture [10,11] (Figure 1). 

### 3.2. Available Formulations

AZT is available in both oral and parenteral formulation and, by a pharmacokinetic point of view, it shows good oral bioavailability (64%) with serum concentration peak 30 to 90 min after administration [12]. The AZT serum half-life is 1 h but the intracellular half-life is about 3 h and is eliminated with urine after undergoing glucuronidation through the liver. Of note, AZT good distribution volume allows good penetration in the central nervous system (CNS) and through the placenta [12].

### 3.3. Recommended Doses

AZT is recommended at the dosage of 300 mg every 12 h in oral formulation and 1 mg/kg/dose every 4 h in the prevention of perinatal transmission of HIV (2 mg/kg loading dose followed by continuous infusion of 1 mg/kg until delivery).

### 3.4. Place in Therapy

AZT use is no longer recommended in the treatment of HIV infection; the role is limited to prevent mother-to child transmission: (a) intrapartum in women with HIV-RNA > 50 copies/mL, and (b) in infants of HIV-infected mothers [13,14] and as deep salvage treatment in patients with proven sensitivity at the genotypic resistance test [15]. Side effects, in particular severe anemia, gastrointestinal toxicity and early viral resistance, were the main barriers to usage.

## 4. Enfuvirtide

### 4.1. Molecular Characteristics

Enfuvirtide (T-20) was the first antiretroviral drug targeting viral entry process; it is a 36-aminoacid polypeptide derived from heptad repeat (HR) 2 that binds the HR-1 region of gp41 preventing the coil–coil zipping reaction and subsequently HIV-1 fusion with cellular membrane [16]. T-20, due to its peculiar mechanism of action, is active on both CXCR4-tropic and CCR5-tropic virus (Figure 2). 

### 4.2. Available Formulations

This drug is available only in a subcutaneous compound due to its susceptibility to intestinal peptidases. T-20 has a concentration peak at 4 h and a plasmatic half-life ranging from 2 to 3. 8 h without influence by renal or hepatic function. There are frequent adverse drug reactions, mainly cutaneous in the injection site.

### 4.3. Recommended Doses

The dosage of T-20 is 90 mg every 12 h by subcutaneous administration.

### 4.4. Place in Therapy

T-20 use is no longer recommended and limited to as deep salvage treatment in patients with previous treatment failures. In these individuals, the use is recommended in combination with an optimized background therapy [17,18].

## 5. Rilpivirine (RPV; TMC278) Long Acting (RPV-LA)

### 5.1. Molecular Characteristics

Rilpivirine (RPV or previously TMC278) is a potent diarylpyrimidine NNRTI currently used for the treatment of ART-naïve patients with a CD4+ T cell count more than 200 cells/mm^3^ and HIV-1 RNA levels less than 100,000 copies/mL and of ART-experienced patients without resistance mutations to RPV (Figure 3). 

### 5.2. Available Formulations

It is available in oral formulation (25 mg once daily) as an individual drug preparation or as a single-tablet regimen (STR) in combination with tenofovir disoproxil fumarate or tenofovir alafenamide and emtricitabine (TDF/FTC/RPV; TAF/FTC/RPV) or with dolutegravir (RPV/DTG).

In oral formulation, RPV showed a good tolerability and a good pharmacokinetic (PK) profile. RPV should be taken with a normal caloric meal (533 Kcal) or high-fat high-caloric meal (928 Kcal) and need careful evaluation of possible drug interactions, in particular with proton pump inhibitors and antiepileptic drugs. In a trial on healthy volunteers that were exposed to intravenous nanosuspension, maximum concentration was achieved in three days and decreased to below 10 ng/mL after 12–26 weeks with a half-life of five weeks [19,20,21].

### 5.3. Recommended Doses

In animal models, RPV, formulated as a 200 nm nanosuspension and administrated intramuscularly (IM) or subcutaneously (SC), showed sustained and dose-proportional release over 2–6 months with an absolute bioavailability of 100% in spite of high concentrations at the injection site. IM administration showed a higher peak concentration compared to SC. This study showed that nanoparticles generated a secondary depot of RPV by macrophage in the lymph nodes with an initial concentration 100-fold higher than plasma one month after the administration, which decreased 3–6 folds below plasma concentration beyond three months [22].

Long-acting RPV (RPV-LA) was also evaluated in a phase I study with a single injection of escalating doses and a phase II study with 1200 mg loading dose followed by 600 mg every four weeks. RPV was detected after 4 and 8 h after administration; analysis of plasma concentrations was performed up to 28 days after injection or up to <20 ng/mL. After a 300-, 600- or 1200-mg injection, RPV-LA PK was largely comparable to the 1200-mg loading dose and both 600-mg RPV-LA injections. The mean plasma concentration of RPV when administered every 28 days after the last injection in all dosing groups was 79 ng/mL [23].

#### Clinical Trials with Cabotegravir (CAB-LA)

1. The LATTE-2 study evaluated cabotegravir and RPV-LA in maintaining HIV-1 viral suppression for 96 weeks. In the randomized, phase 2b, open-label study, HIV-1-treated adults initially received oral cabotegravir (CAB) plus ABC/3TC once daily. The objective of this study was to select an effective, tolerable, and safe intramuscular dosing regimen of the two drugs compared with oral CAB plus ABC/3TC. After a 20-week induction period, patients with HIV-1 RNA <50 copies per mL were randomly assigned (2:2:1) to therapy with CAB LA + RPV-LA at 4-week intervals or at 8-week intervals or to continue oral CAB plus ABC/3TC. The primary endpoints were the proportion of patients with viral suppression at week 32, protocol-defined virologic failures, and safety events for 96 weeks. A total of 309 patients were enrolled, 286 were randomly assigned to the maintenance period (115 to each of the 4- and 8-week groups and 56 to the oral treatment group). At week 96, viral suppression was maintained in 47 (84%) of 56 patients receiving oral treatment, 100 (87%) of 115 patients in the 4-week group, and 108 (94%) of 115 patients in the 8-week group. Three patients (1%) had virological failure (two in the 8-week group; one in the oral treatment group). Injection site reactions were mild (3648 (84%) of 4360 injections) or moderate (673 (15%) of 4360 injections) and rarely led to discontinuation (two (<1%) of 230 patients); injection site pain was most frequently reported. Serious adverse events during maintenance were reported in 22 (10%) of 230 patients in the intramuscular groups (4- and 8-week groups) and in seven (13%) of 56 patients in the oral treatment group; none were drug-related. In this study, the combination of CAB-LA and RPV-LA every four weeks or every eight weeks was as effective as daily three-drug oral therapy in maintaining HIV-1 viral suppression for 96 weeks and was well accepted and tolerated [24].

2. In a phase 3, randomized, open-label trial ART-treatment, naïve patients were given 20 weeks of daily oral induction therapy with DTG-abacavir (ABC)-lamivudine (3TC) (DTG/ABC/3TC). Participants who had an HIV-1 RNA <50 copies/mL after 16 weeks were randomly assigned (1:1) to continue the DTG/ABC/3TC or switch to oral cabotegravir plus RPV for one month followed by monthly injections of long-acting cabotegravir plus RPV. The primary end point was the percentage of participants who had an HIV-1 RNA >50 copies/mL or more at week 48 (Food and Drug Administration snapshot algorithm). At week 48, an HIV-1 RNA >50 copies/mL or more was found in 6 of 283 participants (2.1%) who received LA therapy and in 7 of 283 (2.5%) who received oral therapy. This exciting result met the noninferiority criterion for the primary end point (margin, six percentage points). Regarding adverse reactions, in the group of participants who received LA, 86% reported injection site reactions (median duration, 3 days; mild or moderate severity, 99% of cases); 4 participants withdrew from the trial for injection-related reasons. Treatment satisfaction increased after participants switched to LA therapy; 91% preferred long-acting therapy at week 48 [25].

3. In a phase 3, open-label, multicenter, noninferiority study involving patients with HIV-RNA <50 copies/mL for at least six months while taking oral ART, they were randomized into two groups (1:1) to continue their oral therapy or switch to RPV-LA and cabotegravir-LA. The primary end point was the percentage of participants with an HIV-1 RNA level of 50 copies/mL or greater at week 48.

Treatment was initiated in 308 participants per group. At week 48, HIV-1 RNA levels of 50 copies/mL or more were detected in five participants (1.6%) on long-acting therapy and in three (1.0%) on oral therapy (adjusted difference, 0.6 percentage points; 95% confidence interval [CI], −1.2 to 2.5), a result that met the noninferiority criterion for the primary end point (noninferiority margin, six percentage points). Adverse events were more common in the long-acting group and included injection pain, which occurred in 231 patients (75%), however, mild or moderate in most cases; and 1% withdrew because of this event [26].

### 5.4. Place in Therapy

The combination of RPV-LA and CAB-LA is recommended for maintenance treatment of HIV-infected adults who have HIV-RNA < 50 copies/mL, and when the virus has not developed resistance to NNRTIs and integrase strand transfer inhibitor (INIs).

### 5.5. Place in Pre-Exposure Prophylaxis

RPV LA IM injections every eight weeks in the HPTN 076 study were safe and acceptable. Overall, despite more injection site reactions and pain in the participants receiving RPV LA, the injections were well tolerated. Data from this study support the further development of injectable PrEP agents [27].

## 6. Cabotegravir (CAB, GSK 1265744)

### Molecular Characteristics and Available Formulations

Cabotegravir (CAB) is an integrase strand transfer inhibitor (InSTI) similar to DTG that was initially studied in oral formulation (Figure 4). Then, it was made available in subcutaneous and intramuscular formulations using nanotechnology, achieving a long half-life, which allows for a monthly or quarterly dosing schedule with a good safety profile [28].

## 7. Clinical Trials with RPV (RPV-LA)

See studies previously described in RPV-LA

### 7.1. Place in Therapy

The combination of RPV-LA and CAB-LA is recommended for maintenance treatment of HIV-infected adults who have HIV-RNA < 50 copies/mL, and when the virus has not developed resistance to NNRTIs and InSTIs.

### 7.2. Place in Pre-Exposure Prophylaxis

Recently, the HIV Prevention Trials Network study (HPTN 084) on the safety and efficacy of CAB LA for pre-exposure prophylaxis (PrEP) in HIV-uninfected women, was stopped early by the trial Data and Safety Monitoring Board (DSMB) as results showed CAB LA to be highly effective in preventing HIV acquisition [29].

## 8. Ibalizumab

### 8.1. Molecular Characteristics

Ibalizumab (IBA) (Trogarzo™), formerly known as TNX-355, is a humanized IgG4 monoclonal antibody (MAb), which acts as an entry inhibitor. It binds the CD4 receptor, leading to its conformational change and interfering with its binding with HIV gp120 [30,31] (Figure 5). 

### 8.2. Available Formulations

IBA is currently available as an intravenous infusion and, given its long half-life (3–3.5 days after both intravenous and subcutaneous administration), it can be administered every two weeks as a single loading dose of 2000 mg, followed by a maintenance dose of 800 mg.

This first biologic drug was studied for more than a decade with phase I and II trials demonstrating its efficacy and good tolerability profile.

### 8.3. Recommended Doses

In a single-group, open label, phase 3 study, 40 patients with an HIV-RNA higher than 1000 copies/mL, ART-treated for more than six months with a 3-class resistance were enrolled. The primary endpoint was the proportion of PLWH with a decrease in HIV-RNA load of at least 0.5 log_10_ copies/mL from baseline (day 14). A total of 31 PLWH completed the study. Of the 40 PLWH in the ITT population, 33 (83%) had a decrease in HIV-RNA load of at least 0.5 log_10_ copies/mL from baseline. At week 25, a mean VL decrease of 1.6 log_10_ from baseline was observed. Undetectable VL was observed in 43% of patients, while 50% of patients had a VL < 200 copies/mL. The most common adverse events were diarrhea; 4 patients died from causes related to underlying illnesses while 1 patient developed an immune reconstitution syndrome [31].

### 8.4. Place in Therapy

IBA was approved by the FDA in 2018 and by the EMA on 26 September 2019 [32] in combination with other antiretroviral(s) and is indicated for the treatment of adults infected with multidrug resistant HIV-1 infection for whom it is otherwise not possible to construct a suppressive antiviral regimen.

## 9. GS-6207

### 9.1. Molecular Characteristics

GS-6207 is a first-in-class inhibitor of HIV-1 capsid function (CA) with a long-acting activity that can be administered monthly or less frequently in extended-release parenteral formulations thanks to its low aqueous solubility. It binds with high affinity to CA hexamers between adjacent CA monomers and retains sensibility against HIV-1 mutants resistant to other ARV classes including mutants with a Gag alteration that conferred resistance to other maturation agents (Figure 6).

In an in-vitro study, it displayed potent activity in MT-4 cells (EC50 = 0.1 nM, CC50 = 27 μM) and exhibited a mean EC50 of 0.05 nM in human PBMCs against 23 HIV-1 clinical isolates including all subtypes. In the same study, GS-6207 human serum protein-adjusted EC95 was found to be >10-fold lower than that of efavirenz (EFV), RPV, DTG, and atazanavir (ATV) while in primary human CD4+ T-cells and macrophages was >10-fold more potent and >22-fold more selective than them. GS-6207 also suppressed HIV-2 replication. GS-6207 antiviral activity decreased with increasing multiplicity of infection (MOI), but remains 5- to >100-fold more potent than four commonly used ARVs and exhibits low cytotoxicity in four human cell lines and primary hepatocytes (CC50 > 44 μM). It also showed synergistic antiviral activity when combined with other ARV classes [33].

GS-6207 disrupts HIV capsid, a multimeric shell that is essential to viral replication, at multiple stages throughout the viral life cycle.

Results from a phase 1b proof-of-concept study of a subcutaneous formulation showed antiviral activity with GS-6207 with significantly greater reductions in HIV-1 RNA versus the placebo across all treatment groups.

Preliminary PK data were consistent with sustained delivery with a dose interval of at least three months with a median apparent terminal t_1/2_ between 30 to 38 days and concentrations measurable for at least 16 weeks. Tmax values ranged from 21 to 35 days and the increase in exposure (C_max_ and AUC) between 30 and 100 mg was proportionate to dose [34].

### 9.2. Place in Therapy

The above-mentioned data highlight the potential of GS-6207 for treatment in PLWH, regardless of ARV history.

## 10. Albuvirtide (ABT)

### 10.1. Molecular Characteristics

Albuvirtide (ABT) is a new injectable fusion inhibitor with once-a-week administration currently approved in China (Figure 7).

ABT is a 3-maleimidopropionic acid (MPA)-modified peptide HIV fusion inhibitor derived from the N-terminal sequence of HIV-1 gp41 which can irreversibly conjugate to serum albumin that extends its half-life. Unlike enfuvirtide, it manages to form a stable α-helical conformation with the target sequence of gp41 and it blocks the fusion-active six-helix bundle (6-HB) formation in a dominant-negative manner preventing viral fusion and entry [35]. This mechanism of action permits potent inhibitory activity of ABT against a broad spectrum of HIV-1 strains including variants resistant to T20 [36].

This molecule was evaluated in vivo by Zhang et al. in an open-label and randomized trial on 20 ART naïve patients with a HIV-RNA > 5000 copies/mL and a CD4+ cell count >350 cells/mm^3^. They received either an intravenous weekly infusion of 160 or 320 mg of ABT and lopinavir/ritonavir (LPV/r) 400/100 mg twice daily. At week 7, decline of 1.9 (1.3–2.3) log_10_ and 2.2 (1.6–2.7) log_10_ copies/mL of HIV-1 RNA was observed from the baseline. Viral load suppression (HIV-RNA < 50 copies/mL) was achieved in 11.1% (1/9) and 55.6% (5/9) patients, while the CD4+ cell count change was −5 cells/mm^3^ and 52 for the 160 and 320 mg dose group, respectively [37].

Results from the phase 3 multicenter, open-label, randomized, controlled, non-inferiority TALENT (Test ALbuvirtide in experiENced patienTs) in HIV-1 infected patients who failed ART were presented. Adults infected with HIV-1 were eligible for inclusion if they had a plasma viral RNA load of at least 1000 copies/mL or higher. Patients were allocated (1:1) to the ABT plus lopinavir boosted with the ritonavir (LPV/r) group or the LPV/r plus two optimized NRTIs groups were enrolled with 347 PLWH for 48 weeks. A HIV-RNA load <50 copies/mL was obtained in 66% of the 2 NRTI group and 80% of the ABT group reaching the non-inferiority endpoint. Only in five patients in the ABT arm was a HIV-RNA load >500 copies/mL detected without evidence of resistance. No death occurred in the two groups and the more common adverse events were gastrointestinal abnormalities, rash, headache, dizziness, hematuria, cholesterol, and triglyceride elevation with no significant differences between the two groups [38].

### 10.2. Place in Therapy

Currently, its role in therapy is still to be defined, potentially in heavily treatment experienced patients with limited treatment options left.

## 11. bNAbs

HIV-1-specific broadly neutralizing antibodies (bNAbs) may be a key player as an “add-on” agent for HIV prevention, treatment, and perhaps, eradication [39].

bNAbs have been shown to reduce VL and to potentially maintain viral suppression; however, rebound viremia with emergence of resistant acid strains has been observed in clinical trials as well as a not fully understood effect of the viral reservoir [39].

### Place in Therapy

Although efficient neutralization of free HIV virus has been described, bNAbs need further investigations to find their proper place in therapy.

## 12. Discussion and Conclusions

In the last few years, much progress has been made in terms of antiretroviral therapy regimens with new drugs and new formulations, but HIV-resistant strains, ART interactions, and long-term ART adherence remain important issues [40,41,42,43].

The availability of routes of administration other than oral such as subcutaneous or intravenous may be an advantage and may allow the physicians to closely monitor assumption with direct observed therapy, especially in populations at high risk for interruptions (e.g., adolescents) or in PLWH who report drug fatigue. Long-acting drugs with administration every 1–2 or more months may effectively reduce poor adherence to treatment.

Pharmacokinetics and pharmacodynamics of injectable drugs allow more regular drug level through adequate therapeutic drug monitoring (TDM), decreasing the likelihood of erratic kinetics due to oral administration (e.g., with or without meals, gastric pH level), and a more stable antiretroviral effect in an optic of treat as prevention (TaSP) and PrEP.

Not the least, the access to molecules with a new mechanism of action opens awaited therapeutic chances to those heavily-experienced patients with few or no available treatment options (e.g., Ibalizumab).

In conclusion, injectable agents might represent a turning point in the care of people living with HIV, allowing less frequent administration of ART, and hopefully more widespread use of PrEP, more stable drug levels in the blood, thus increasing the odds to get closer to ending the HIV epidemic.

## Figures and Tables

**Figure 1 viruses-13-00228-f001:**
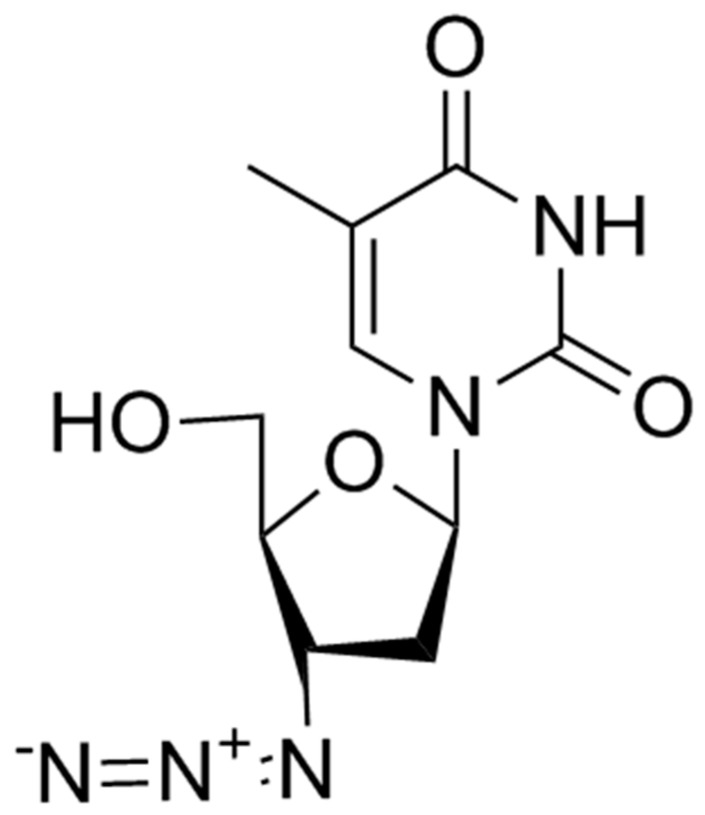
Zidovudine.

**Figure 2 viruses-13-00228-f002:**
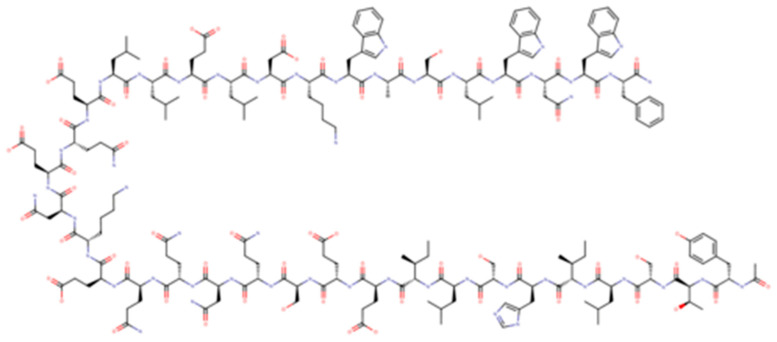
Enfuvirtide.

**Figure 3 viruses-13-00228-f003:**
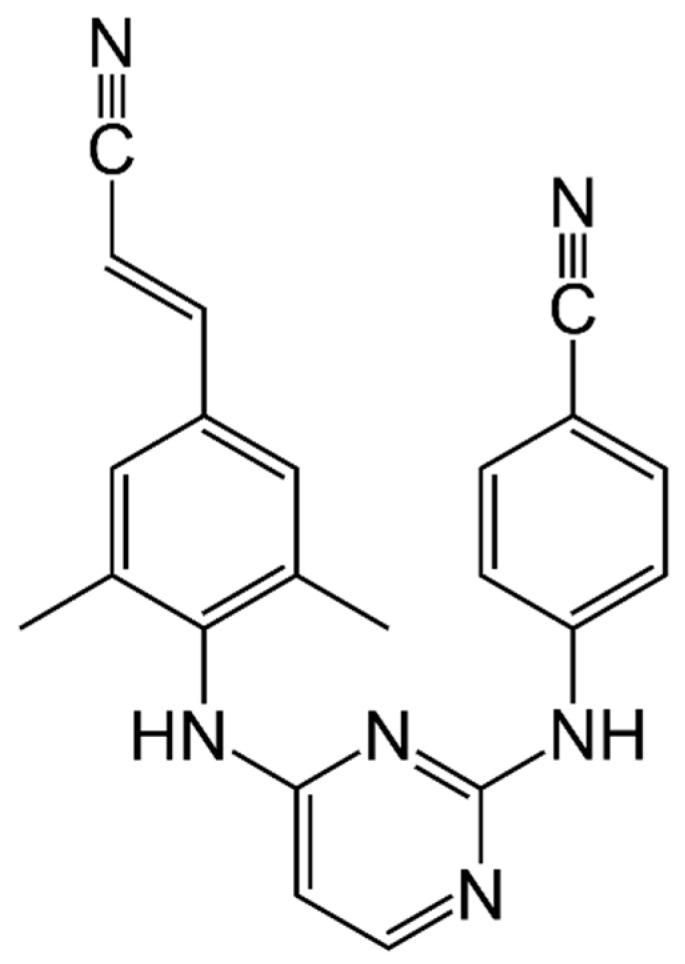
Rilpivirine.

**Figure 4 viruses-13-00228-f004:**
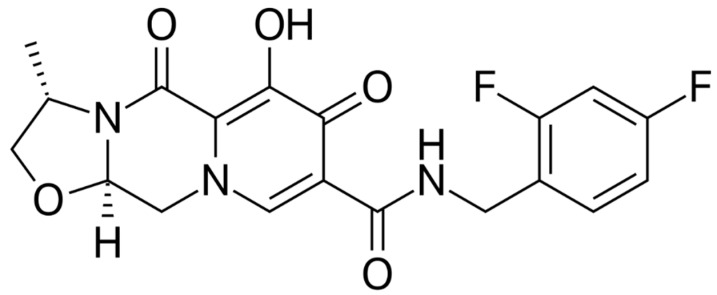
Cabotegravir.

**Figure 5 viruses-13-00228-f005:**
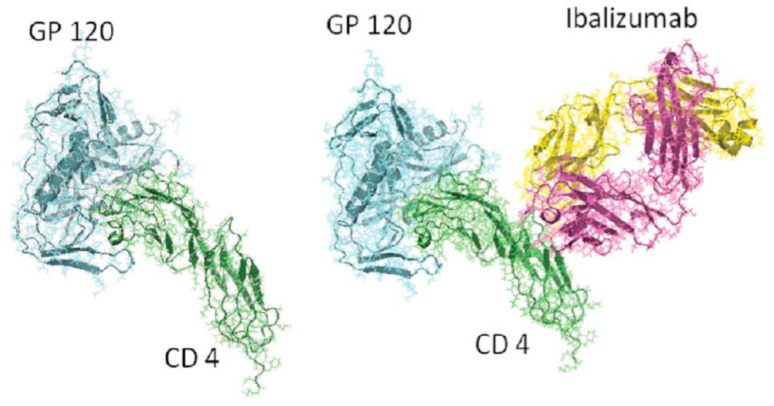
Ibalizumab.

**Figure 6 viruses-13-00228-f006:**
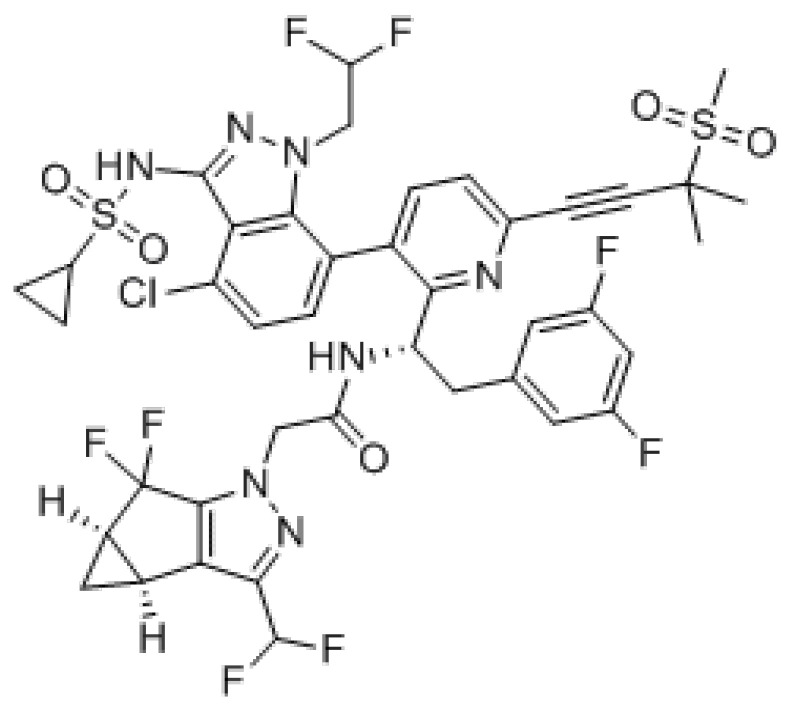
GS-6207.

**Figure 7 viruses-13-00228-f007:**
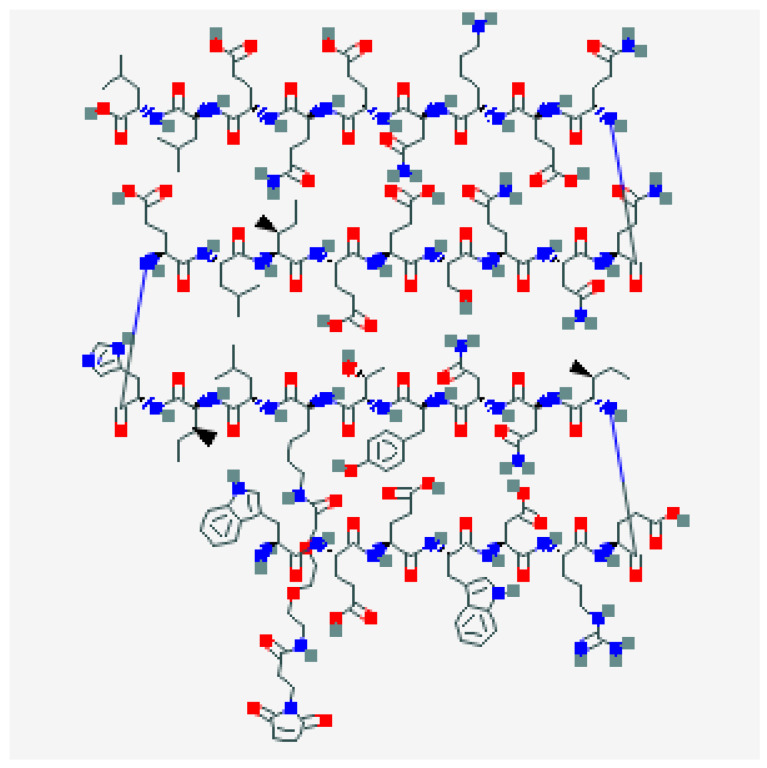
Albuvitirde.

## Data Availability

Data sharing not applicable.

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
