# Peer review of "Injectable Antiretroviral Drugs: Back to the Future"

_viruses, 2021, doi:10.3390/v13020228_

Round 1

Reviewer 1 Report

Injectable Antiretroviral Drugs: Back to the Future

Marco Berutti et al

This is presented as systematic review of injectable antiretroviral drugs with stringent Pubmed search criteria. The review describes and present a number of individual compounds with regard to molecular mode of action, structure and formulation.

I had initial expectations to this review as much progress and exciting developments have been made in novel antiretroviral drug development both with advent of novel drug classes and point of attack and in the area of long acting formulation (of which many are injectables).

The review need some major additional work to become relevant today.

First the search of the literature ended in March 2019??? Why not updated to today November 2020. By simple drug search many highly relevant papers have been identified.

Orkin et al NEJM March 2020

Swindell et al NEJM March 2020

Of high relevance to Rilpivirine and there are others relevant to include and discuss in the clinical utility of these exciting drugs

For GS-6207 there are also several papers published for example

Yant et al Nat Medicine 2019

Link et al Nature 2020

And for Cabotegravir the same is true.

On page 6 line 210-212: “Although the clinical trials FLAIR (NCT02938520) and ATLAS (NCT02951052) are still on-going, preliminary results showed that cabotegravir LA and rilpivirine LA based regimens could become an interesting therapeutic option in selected naïve and ART-experienced patients.”

These trials have ended and have been reported!!!!

The authors have no sections on broadly neutralizing antibodies (which are injectables) which is a very interesting novel drug class with great potential. There are numerous papers with clinical data on both the antibodies from the Nussenzweig lab and from the VRC development. Also published before March 2019. This need to be included as there are great prospects for both treatment and prevention.

And these exciting developments are neglected but the review starts with a presentation of two inferior and today clinically obsolete drugs AZT and Enfuvirtide. Why are these included at all?

Lastly in the section on Ibalizumab the authors have contrary statements right after each other. Page 7, line 233-237:

Currently TaiMed Biologics is actively engaged with the EMA to pursue the regulatory pathway for ibalizumab in the European Union [31].

Place in therapy

IBA was approved by FDA in 2018 and by EMA in September 26, 2019 for the treatment of heavily treatment-experienced adults with multidrug resistant HIV-1…”

There are minor issues with the stringency of paragraphs for some drugs a detailed sub-breakdown in molecular characteristics, formulations and dosages and place in therapy. Some drugs have very detailed trial reporting others have no outcome reporting.

Please make it stringent and coherent.

The references are mismanaged. Several reference numbers for single references (ie ref 9-16 and ref 38-41 and 42-45). Further ref 38 is in the section of GS-6207 but is not related to GS-6207

Author Response

Dear Editor,

Thank you for your kind assessment of our manuscript and for the suggestions that helped us to improve the quality of the content.

We are here enclosing a point-by-point response to the Reviewers’ comments, questions, and remarks. Overall, the Reviewers raised relevant issues and we are grateful for their time and efforts.

Please find in the reviewed manuscript all the corrections done after the revision. All the modifications are highlighted in yellow.

Reviewer 2 Report

Please, review acronyms and units throughout the manuscript, such as PrEP (not PreP), HIV RNA levels in copies/mL, and for CD4+ T cell count use cells/mL or cells/mm3.

Review references numbering since they do not match with the numbers in the text.

Line 27: HIV “pandemic” instead of “epidemic.”

Line 25: Anti-retroviral is one word “antiretroviral.”

Line 24: People with HIV (PWH).

Line 34: Cases of HIV were reported.

Line 38: Sub-Saharan Africa instead of Sub-Saharan area.

Line 39: constant threat to achieve the 90-90-90.

Line 46: The aim of this manuscript …

Lines 65-66: review zidovudine mechanism of action. From package insert:

“Mechanism of Action: Zidovudine is a synthetic nucleoside analogue. Intracellularly, zidovudine is phosphorylated to its active 5′-triphosphate metabolite, zidovudine triphosphate (ZDV-TP). The principal mode of action of ZDV-TP is inhibition of reverse transcriptase (RT) via DNA chain termination after incorporation of the nucleotide analogue. ZDV-TP is a weak inhibitor of the cellular DNA polymerases α and γ and has been reported to be incorporated into the DNA of cells in culture.”

Line 83: review AZT DDI. The are other ARV drugs with higher number of DDIs than AZT.

Line 84: review grammar.

Line 85: remove “post-exposure prophylaxis” since it may be confusing.

Line 86: review this sentence. AZT indication for MTCT of HIV.

Line 111: review sentence. It should read: ARV-treatment naïve patients with CD4+ T cell count more than 200 cells/mm3 and HIV-1 RNA levels less than 100,000 copies/mL.

Line 120: rilpivirine is should be taken with a normal caloric meal (533 kcal) or high-fat high-caloric meal (928 kcal).

Line 167: the abstract states that the switch was after 20 and not 96 weeks: “After a 20-week induction period on oral cabotegravir plus abacavir–lamivudine, patients with viral suppression (plasma HIV-1 RNA <50 copies per mL) were randomly assigned (2:2:1) to intramuscular long-acting cabotegravir plus rilpivirine at 4-week intervals (long-acting cabotegravir 400 mg plus rilpivirine 600 mg; two 2 mL injections) or 8-week intervals (long-acting cabotegravir 600 mg plus rilpivirine 900 mg; two 3 mL injections) or continued oral cabotegravir plus abacavir–lamivudine.”

Line 183: FLAIR has been already published in N Engl J Med 2020; 382:1124-1135. Please, update data and reference.

Line 196: ATLAS has been already published in N Engl J Med 2020; 382:1112-1123. Please, update data and reference.

Line 222: if the compound is only available by IV infusion, why the half-life is given for the SC administration?

Line 233: the following sentence states that the EMA has already approved it on September 2019.  Reivew this sentence.

Line 227: update Ibalizumab data published in N Engl J Med 2018; 379:645-654. Update reference.

Line 239: Islatravir is an oral formulation. Implants are in developments.  It is not injectable.

Line 282: annual implants are under evaluation. It is not an “available formulation.”

Line 291: review the meaning of “prophylaxis” in the “prophylaxis and PrEP.”

Line 293: GS-6207, update with data presented at the Conference on Retroviruses and Opportunistic Infections (CROI) 2020 in Boston.

Line 344: Update data and reference for TALENT study published in Chinese Medical Journal: December 20, 2020 - Volume 133 - Issue 24 - p 2919-2927.

Line 360: dysphagic or unconscious patients are not the main causes of non-adherence to ARV therapy.  The main benefit is on long-acting drugs with administrations every 1, 2 or more months to improve adherence.

Author Response

(The authors gave the same response as above.)

Round 2

Reviewer 1 Report

My only concern remain the absence of broadly neutralising antibodies. The authors state it should be reviewed elsewhere but I fail to see why.

It may be the longest acting injectable in the pipeline

Author Response

Dear Editor,

Thank you for your kind assessment of our manuscript and for the suggestions that helped us to improve the quality of the content.

We are here enclosing a point-by-point response to the Reviewer’s comment.

Please find in the reviewed manuscript all the corrections done after the revision. All the modifications are highlighted in yellow.

#Reviewer 2

My only concern remain the absence of broadly neutralising antibodies. The authors state it should be reviewed elsewhere but I fail to see why. It may be the longest acting injectable in the pipeline

Authors’ Answer

As suggested by the Reviewer, we updated the manuscript with a short paragraph regarding broadly neutralizing antibody.
